# Eigenvector alignment: Assessing functional network changes in amnestic mild cognitive impairment and Alzheimer's disease

**Ruaridh A. Clark** [1]*, **Niia Nikolova**[2], **William J. McGeown** [2], **Malcolm Macdonald**[1]

**1** Electronic and Electrical Engineering, University of Strathclyde, Glasgow, United Kingdom, **2** School of Psychological Sciences and Health, University of Strathclyde, Glasgow, United Kingdom

* ruaridh.clark@strath.ac.uk

**Data Availability Statement:** All 'Resting-state fMRI in Dementia Patients' subjects files are available from the Harvard Dataverse database (doi:10.7910/DVN/29352).

## Abstract

Eigenvector alignment, introduced herein to investigate human brain functional networks, is adapted from methods developed to detect influential nodes and communities in networked systems. It is used to identify differences in the brain networks of subjects with Alzheimer's disease (AD), amnestic Mild Cognitive Impairment (aMCI) and healthy controls (HC). Well-established methods exist for analysing connectivity networks composed of brain regions, including the widespread use of centrality metrics such as eigenvector centrality. However, these metrics provide only limited information on the relationship between regions, with this understanding often sought by comparing the strength of pairwise functional connectivity. Our holistic approach, eigenvector alignment, considers the impact of all functional connectivity changes before assessing the strength of the functional relationship, i.e. alignment, between any two regions. This is achieved by comparing the placement of regions in a Euclidean space defined by the network's dominant eigenvectors. Eigenvector alignment recognises the strength of bilateral connectivity in cortical areas of healthy control subjects, but also reveals degradation of this commissural system in those with AD. Surprisingly little structural change is detected for key regions in the Default Mode Network, despite significant declines in the functional connectivity of these regions. In contrast, regions in the auditory cortex display significant alignment changes that begin in aMCI and are the most prominent structural changes for those with AD. Alignment differences between aMCI and AD subjects are detected, including notable changes to the hippocampal regions. These findings suggest eigenvector alignment can play a complementary role, alongside established network analytic approaches, to capture how the brain's functional networks develop and adapt when challenged by disease processes such as AD.

## Introduction

Functional connectivity is one of the gateways through which a network representation of the brain's interactions can be sought. These connectivities can be weighted based on the strength of the correlations between regions over time. This measures may provide information on the

**Funding:** The author(s) received no specific funding for this work.

**Competing interests:** The authors have declared that no competing interests exist.

underlying reasons that clinical symptoms occur at different disease stages. Researchers have used functional connectivity values to assess changes in the strength of interaction between groups of key regions [1, 2]. These methods can provide clear results but do not take the holistic view required to capture how variations in functional connectivity can change the functional networks of the brain. Graph theory and network neuroscience methods exist for evaluating structural network changes [3], with the assessment of functional connectivity of subjects in resting-state a popular field of study [4]. But these graph structural assessments do not explicitly capture the changing relationship between nodes. It is also important to note that these methods often require binarisation and thresholding of the functional connectivity matrices, which are all-to-all, weighted, adjacency matrices. Binarisation produces an unweighted graph, which is potentially an inaccurate representation of a neuronal network since connections are known to vary in strength [5]. Thresholds are frequently applied to remove potentially spurious low weight connections, but their application can be controversial as the choice of threshold value can affect the results [5–7]. To address this problem some unbiased thresholds have been proposed, such as the Cluster-Span Threshold [8].

This paper focuses on Alzheimer's disease (AD) and amnestic Mild Cognitive Impairment (aMCI). AD is the most common type of dementia. It is typically characterised by a marked decline in episodic memory, with deficits occurring in other cognitive domains such as in language, visuospatial and executive functioning. Individuals with aMCI present with impairments in memory, but their other cognitive domains remain relatively intact. Not all individuals with aMCI are in the early stages of AD, but people with this pattern of symptomology are at high risk of conversion to the disorder [9]. There has been a large body of work documenting the changes in regional brain volume as AD progresses, for example, individuals with aMCI that later convert to Alzheimer's disease have been shown to have greater atrophy of the left hippocampus than those who do not [10]. Volumetric analysis is useful but it does not inform on how brain dynamics are modified by AD, with *functional magnetic resonance imaging* (fMRI), the data is able to offer information on the regulation of brain networks that can provide additional markers of disease. Decreased functional connectivity has also been proposed as a biomarker for AD [11].

The theoretical basis for eigenvector alignment, proposed in this paper as a tool for detecting and quantifying changes in functional alignment, is a method of influential community detection, referred to as communities of dynamical influence (CDI) [12]. Communities are usually detected by an increased density of connections, but for CDI this density is implicitly captured by proximity and alignment to the network's most influential nodes. Communities have been detected in a similar manner before, with [13] also employing eigenvectors to define divisions in the network structure, and CDI itself has been applied in the same context of comparing AD and aMCI subjects [14]. But by breaking CDI down into its two main components, which are eigenvector centrality and eigenvector alignment, we can gain insights into the shifting relationships between brain regions. Eigenvector centrality is an established metric in network science for determining the influence of a network node in terms of the flow of information around a system [15, 16]. Eigenvector alignment is presented herein as a method to determine the alignment of Regions of Interest (ROIs) when they are embedded in Euclidean space defined by the system's dominant eigenvectors. Eigenvectors capture every change in functional connectivity to provide an insight into functional network changes across the whole brain, with eigenvector alignment able to identify the impact on any two ROIs. Interestingly, our approach draws strong parallels to [17, 18], where a measure of functional connectivity itself is derived by converting ROIs from an anatomical space to a functional space (or eigenimage).

Eigenvectors have been used previously to identify circuitry in neuronal networks, where [19] essentially applied spectral bisection but using a selection of eigenvectors, not just the second dominant eigenvector, to reveal known associations of neurons in a *C. elegans* neuronal network. An expansion of this approach for identifying neuronal circuitry, but using multiple eigenvectors in combination, has also been presented in [20].

Eigenvector alignment is applied throughout this paper using three dominant eigenvectors, which is suggested in [12] as a trade-off between capturing community structure from the network and ensuring that those communities represent the most important divisions in terms of information flow dynamics. The most dominant eigenvectors are associated with the eigenvalues of largest magnitude. When treating the adjacency matrix as a linear transformation these eigenvectors capture the direction of greatest linear change for this transformation. Hence, these vectors highlight the ROIs that are most important for the movement of information around the network and likely to be the most critical if disrupted.

## Materials and methods

In order to compare the functional connectivity between the AD, aMCI and Healthy Control (HC) groups, a connectivity matrix is generated for each subject from their resting-state fMRI scan. The connectivity is only considered between a series of predefined brain regions, each defined as a ROI. This results in an all-to-all, weighted and undirected connectivity matrix that captures the strength of the functional connectivity between ROIs. A threshold is applied to reduce the weakest connections and the dominant eigenvectors are calculated for each matrix. These eigenvectors then form the basis for comparing subjects within and between the AD, aMCI and HC groups.

### Dataset

The fMRI resting state data that is analysed in this work is from the 'Resting-state fMRI in Dementia Patients' dataset [21] (Harvard Dataverse). The MRI data was obtained using a Siemens 3T MRI system (Magnetom Allegra, Siemens, Erlangen, Germany) for ten patients with a probable AD diagnosis, 10 aMCI patients [22] and 10 healthy elderly subjects (HC). Probable AD diagnosis was defined by NINCDS-ADRDA consensus criteria [23], with a general cognitive evaluation made using Mini-Mental State Examination (MMSE). The mean MMSE score was 21.5 (standard deviation, SD, 3.7) for the AD group, 25.8 (SD 2.3) for the aMCI group and 29.3 (SD 0.67) for the HC group. The mean age of the AD group was 72.3 (SD 8.3), the mean of the MCI group was 70.7 (SD 7.1) and the mean of the HC group was 66.0 (SD 9.6). The mean education was 8.6 (SD 3.6) in the AD group, 11.1 (SD 3.5) in the MCI group and 14.5 (SD 3.0) in the HC group. There were 6 females in the AD group, 4 in the MCI group and 3 in the HC group. For additional details on the participants in the dataset, see [24].

The subjects underwent a resting state echo-planar imaging (EPI) fMRI scan (TR = 2080 ms, TE = 30 ms, 32 axial slices parallel to AC-PC plane, matrix $64 \times 64$, in plane resolution $3 \times 3mm^2$, slice thickness = 2.5 mm, 50% skip, flip angle = 70 degrees). The duration of the scan was 7 minutes and 20 seconds, yielding 220 volumes. Subjects were instructed to keep their eyes closed throughout, refrain from thinking of anything in particular and to avoid falling asleep. An anatomical T1-weighted three dimensional MDEFT (modified driven equilibrium Fourier transform) scan was also acquired for each subject (TR = 1338 ms, TE = 2.4 ms, TI = 910 ms, flip angle = 15 degrees, matrix = $256 \times 224 \times 176$, FOV = $256 \times 224mm^2$, slice thickness = 1 mm, total scan time = 12 min).

## Preprocessing

The functional data is preprocessed using the CONN toolbox (CONN: functional connectivity toolbox, [25]) for SPM12 (www.fil.ion.ucl.ac.uk/spm) and MATLAB version 2018a.

**Spatial preprocessing.** The first four volumes of the functional scans were removed in order to eliminate any saturation effects and to allow the signal to stabilise. Functional data is slice-time adjusted and corrected for motion. The high resolution T1 weighted anatomical images were coregistered with the mean EPI image. They were segmented into grey matter (GM), white matter (WM), and cerebrospinal fluid (CSF) masks and were spatially normalised to the Montreal Neurological Institute (MNI) space [26]. The obtained transformation parameters were then applied to the motion corrected functional data, and an 8mm FWHM Gaussian kernel was applied for spatial smoothing. It should be noted that the use of spatial smoothing on fMRI data can affect the properties of functional brain networks, including a possible over-emphasis on strong, short-range, links as well as changes in the identities of hubs of the network and decreased inter-subject variation [27].

**Temporal filtering.** The aCompCor technique is applied to mitigate against physiological and movement-related noise. This technique identifies and removes the first five principal components of the signal from the CSF and WM masks (eigenvectors of the PCA decomposition of the EPI timecourse averaged over the CSF and WM), as well as the motion parameters, their first-order temporal derivatives and a linear detrending term [28]. This process resulted in the exclusion of one subject from the AD group due to excessive motion. Scrubbing and motion regression were also performed with the preprocessed functional data then it was bandpass filtered ($0.008\text{Hz} < f < 0.09\text{Hz}$) using a fast Fourier transform (FFT).

## Connectivity matrix generation

The CONN atlas, which combines the FSL Harvard-Oxford cortical and subcortical areas and the AAL atlas cerebellar areas, is used to define one hundred and thirty-two (132) ROIs. Connectivity between the 132 ROIs is assessed for each subject using their 7-minute resting state scan. We constructed 132 x 132 ROI-to-ROI correlation (RRC) matrices of Fisher z-transformed bivariate correlation coefficients (Pearson's r) using the ROIs described above. For each subject, a graph adjacency matrix $A(i, j)$ is computed by thresholding the RRC matrix $r(i, j)$ using the Cluster-Span Threshold (CST [8]). The adjacency matrices, prior to thresholding, are available at [29].

## Cluster-Span Threshold

An unbiased Cluster-Span Threshold (CST) [8] is used for generating the adjacency matrix. CST is especially suitable as it is able to distinguish between HC and AD subjects when applied to their functional connectivity matrices [30]. The threshold is defined based on a clustering coefficient, $C$, that for a given network balances the number of triples that are clustered, forming loops, with those that are spanning, forming trees. A triple is formed by at least two edges connecting 3 nodes. Triples that are also clustered are those that form triangles where each of the three nodes are connected to the other two. A triple that does not form a triangle/cluster is defined as a spanning triple. The threshold generates a topology that excludes edges with weights smaller than the chosen value. CST is selected so that the topology generated contains the same number of clustered triples and spanning triples. The use of thresholds is a common practice to filter out connections that may only be present because of noise in the fMRI data. The choice of threshold can affect the results of a study [5–7], so we briefly compare the use of CST in the Results section with a selection of arbitrary thresholds.

## Communities of dynamical influence

For each subject's connectivity network, the ROIs can be assigned into communities of dynamical influence (CDI) based on the strength and selection of their connections, as described in [12] and available at [31]. CDI identifies communities based on their alignment in Euclidean space defined by multiple (often three) of the system's dominant eigenvectors (those associated with the dominant eigenvalues). The nodes, which are further from the origin of this coordinate system than any of their connections, are defined as leaders of separate communities. Each of these communities is populated with other nodes that lie on a path that connects to the leader node of that community. Each node is assigned to only one community, where the community is chosen by assessing which leader is in closest alignment to that node. This alignment is assessed by comparing the position vector, from the origin of the coordinate system, for the leader nodes with the node to be assigned. The dot product of these position vectors determines the leader that is aligned most closely to that node. The application of CDI on an AD subject's adjacency matrix of functional connectivity is shown in Fig 1, where the network is divided into seven communities.

Once community designation is complete, the order of influence is determined by evaluating the largest entry of the most dominant eigenvector for each community (i.e. eigenvector centrality (EC) [15, 16]) that is known to be a non-negative vector. The community that contains the node with the largest EC value ($\mathbf{v}_1$) is ranked as the most influential community, with the other communities ranked in descending order according to their largest EC value.

In this paper, CDI is determined from the three most dominant eigenvectors of the undirected connectivity matrix after applying CST. These are the eigenvectors associated with the largest eigenvalues in magnitude and are shown in [12] to identify the nodes that are most effective at driving a directed network to consensus. While many other community detection algorithms exist, CDI is the focus here as it explicitly connects network influence with community designation. It is also worth noting that, unlike many other community detection methods, CDI is deterministic and not susceptible to stochastic processes changing community designations.

## Eigenvector alignment

The alignment between every ROI, with respect to every other, can be identified by embedding the ROIs in a Euclidean space, defined by the system's three dominant eigenvectors, and assessing the dot product of their position vectors, with respect to the origin of this coordinate frame. This comparison can yield the angle between the two vectors using the well-known relation,

$$\theta = \cos^{-1}\left(\frac{\mathbf{r} \cdot \mathbf{s}}{|\mathbf{r}||\mathbf{s}|}\right) \tag{1}$$

where $\mathbf{r}$ and $\mathbf{s}$ are the position vectors for different ROIs in the eigenvector based coordinate frame. ROIs are closely aligned when they have a small angle between their position vectors. Also note that position vectors require at least one nonzero entry to produce an alignment angle. In this work any zero length position vector, which occurs if a node has no connections, is given a small value to avoid calculation errors.

The following toy example demonstrates how eigenvector alignment (EA) produces similar results to monitoring changes in functional connectivity (i.e. edge weights) between two ROIs, but differs by considering the network-wide impact of every change in connectivity. In Fig 2, four nodes are connected, with functional connectivity values between 0 and 1, where one edge connecting node B and D has its weight increased from 0.3 to 1. An assessment of

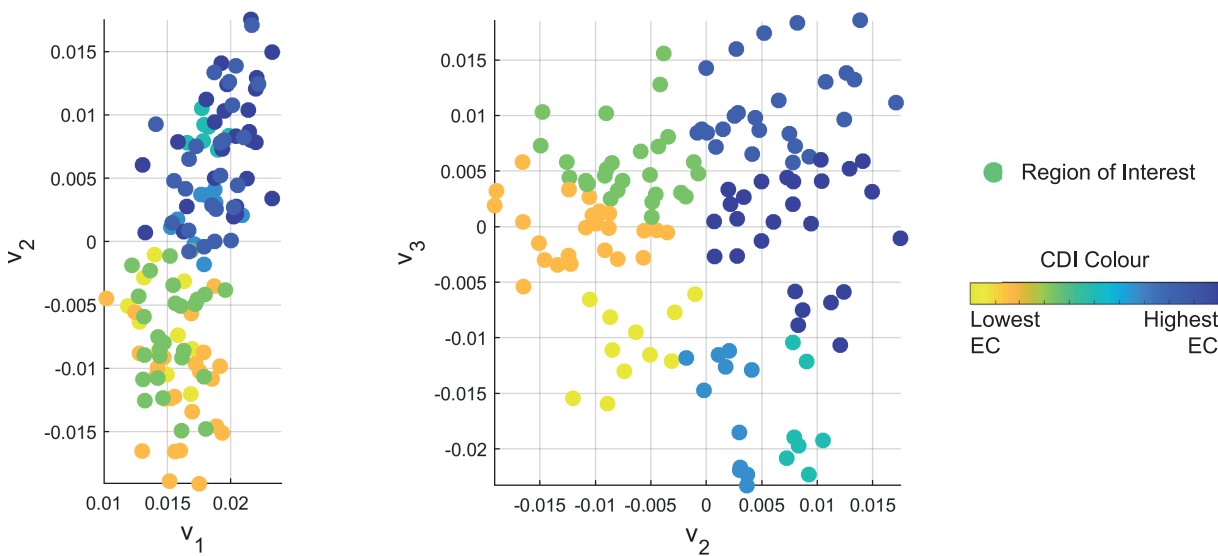

**Fig 1. An AD subject's ROIs embedded in an eigenvector space where $v_1$, $v_2$, and $v_3$ are the three most dominant eigenvectors of the adjacency matrix.** Each community of dynamical influence (CDI) is coloured based on their largest eigenvector centrality (EC) value.

functional connectivity would only identify an increased connection between node B and D. EA also notes that the initial alignment, between node B & D of $\theta$ = 1.57 radians, reduces to $\theta$ = 0.92 rad (i.e. an increase in alignment). However, in contrast to functional connectivity, EA also detects a significant impact to node A as it was initially fully aligned with node D, $\theta$ = 0 rad. After increasing the edge weight, this angle increases to $\theta$ = 0.94 rad, as the node A to D connection is no longer the dominant influence for node D. The other node alignments in this example remain similar, with the next largest change in EA coming from node B becoming less aligned with A (an initial angle of $\theta$ = 1.57 rad increasing to $\theta$ = 1.85 rad).

**Filtering results.** Eigenvector alignment is applied herein to detect differences between pairs of ROIs in functional connectivity networks. The significance of these differences in EA are determined using Welch's $t$ test [32]. To mitigate against false detections of significance, EA is only assessed for ROI pairs that are significant when compared with connectivity matrices generated with uniformly distributed random numbers between 0 and 1. For example, a comparison of the AD and HC groups will only include ROI pairs that display a significant difference in either the AD group or the HC group when compared with sets of 1000 randomly generated functional connectivity matrices. The selection of 1000 random matrices ensures a high consistency of ROI pairs that meet the significance threshold. Comparing between sets of

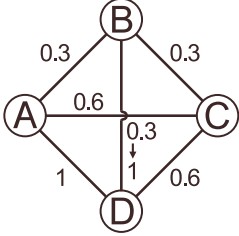

**Fig 2. Toy network example with weighted edges, where the edge weight between node B and D is changed from 0.3 to 1.**

1000 random models produces just over 94% of the same significant ROI pairs, with just over 91% consistently selected by three different sets of 1000 models. Therefore, only the ROI pairs consistently highlighted by all three sets of 1000 models are presented as significant in comparisons with the random models or viable for inclusion in the subject group comparisons.

It is worth noting that a reduction in the number of random models does reduce the consistency of significance detection, but a 200 random model comparison will still produce around a 90% match with the pairs highlighted by a 1000 model. This both demonstrates that EA is capable of consistent findings while emphasises the requirement for filtering on the current dataset where there are at most 10 subjects in a group. The eigenvector alignment algorithm, *t* test scripts and subject dataset are available at [29].

## Results

Cluster-Span Threshold (CST) is applied throughout the following analyses. In [30] clustering coefficient and degree variance were used to demonstrate that CST is a sensitive threshold for AD detection. We demonstrate the effectiveness of CST on this dataset by first dividing the network into communities of dynamical influence (CDI) [12, 31], which are ranked based on their eigenvector centrality as previously described in the Communities of Dynamical Influence section. This ranking captures the influence of communities and enables differences in community size at either end of the influence spectrum to be examined.

When using CST, the number of ROIs in both the most and the least influential communities are seen to significantly vary when comparing the AD subjects with the HC group ($p = 0.011$ and $p = 0.004$ respectively), as depicted in Fig 3 where AD subjects have the largest most influential and the smallest least influential communities. The number of ROIs also vary significantly when comparing the most influential community from the AD and aMCI groups ($p = 0.028$). The finding of significance, in the AD versus HC case, still holds when extending the comparison to include the mean size of the two most and least influential communities, with $p = 0.034$ and $p = 0.014$, respectively, while the AD versus aMCI case falls just outside the threshold $p = 0.053$. The mean number of communities found for all 29 subjects (after thresholding with CST) is around 8 with a minimum of 5 and maximum of 12 communities.

The use of arbitrary thresholds between 0 and 0.9, at 0.1 intervals, rather than CST can also produce significant differences in community size between subject groups. Significant results are recorded for the most influential communities of AD subjects compared to both the HC group (at thresholds 0.2 and 0.3) and the aMCI group (at thresholds 0.1, 0.2, 0.3 and 0.5). While the higher arbitrary thresholds result in significant differences in the sizes of the least influential communities for both the AD subjects compared to the HC group (at thresholds 0.5 and 0.7) and the aMCI group (at threshold 0.7). The CST defined thresholds varied between 0.24 and 0.27 for the 29 subjects, and managed to create a network where both the most and least influential communities differed significantly for AD subjects compared with HC group, as well as finding a significant difference in the AD versus aMCI case. The pattern of AD subjects having larger most influential and smaller least influential communities when compared with HC subjects is resilient to threshold variation.

As noted previously, community structure is linked with network influence when employing CDI. However it is also important to note that CDI is a relatively consistent assessment of network division, as discussed further in the Discussion section, but it can still be susceptible to variation in the relative scaling of eigenvectors and the number of dominant eigenvectors selected for determining communities. For CDI, the first eigenvector is scaled so that its largest entry is equal to that of any of the other eigenvector entries included in the CDI assessment. This ensures that the communities are strongly associated with global network influence,

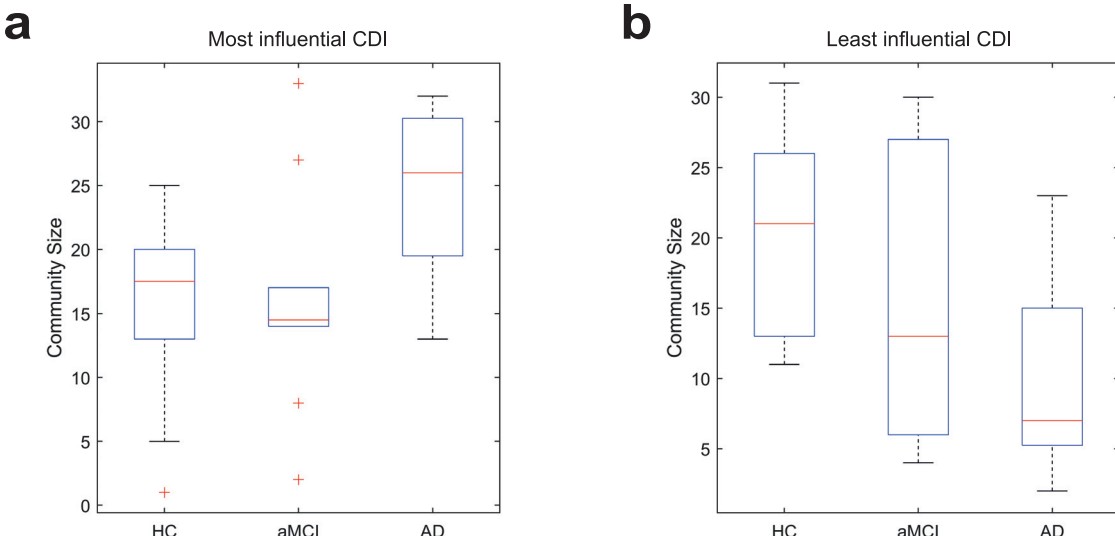

**Fig 3. The variation in community size of (a) the most influential CDI and (b) the least influential CDI for subjects in the HC, aMCI and AD groups.** The median, 25$^{th}$ and 75$^{th}$ percentile are detailed with the whiskers extending to the most extreme data points. Outliers lie more than three scaled median absolute deviations away from the median and are excluded.

rather than local. Within 20% of this scaling there remains a significant difference in the most influential community sizes for the AD and HC comparison, whereas the differences in the AD and aMCI comparison lose significance when increasing the eigenvector entries beyond 10%. The difference in the least influential community comparison between AD and HC groups is much less resilient to change in eigenvector scaling with small changes resulting in loss of significance. Note that the pattern of AD subjects having larger most influential and smaller least influential communities, when compared with HC subjects, is resilient to eigenvector scaling for the assessed range.

In [12], the use of three eigenvectors for CDI was advocated as a trade-off between effective community detection and ensuring communities are determined based on global network influence. The use of three eigenvectors is supported here as the use of two, four or five dominant eigenvectors fails to produce significant differences in community size when applying CST.

## ROI alignment

In the Eigenvector Alignment section, EA is presented as a holistic extension to examining individual functional connectivity changes. To support this claim, we compare the eigenvector alignments from each subject group with those calculated from three sets of 1000 matrices of uniformly distributed random numbers. The significant alignments for each group are presented in Fig 4, where a conservative threshold of $p \leq 2.5 \times 10^{-3}$ was used and the ROI IDs displayed are detailed in S1 Table. These results support the accuracy of EA by noting the large proportion of significant pairs either side of the matrix diagonal. Each of these pairs represent the left and right side of the same cortical area, which as expected would present with an increased alignment when compared with a randomised model of functional connectivity. Not only do these results support the use of EA, in detecting ROI alignment, but they also expose notable differences between the subject groups. In particular, that the AD subjects present with far fewer significant pairs from the same cortical area than the HC group, a finding that will be explored further in the following section.

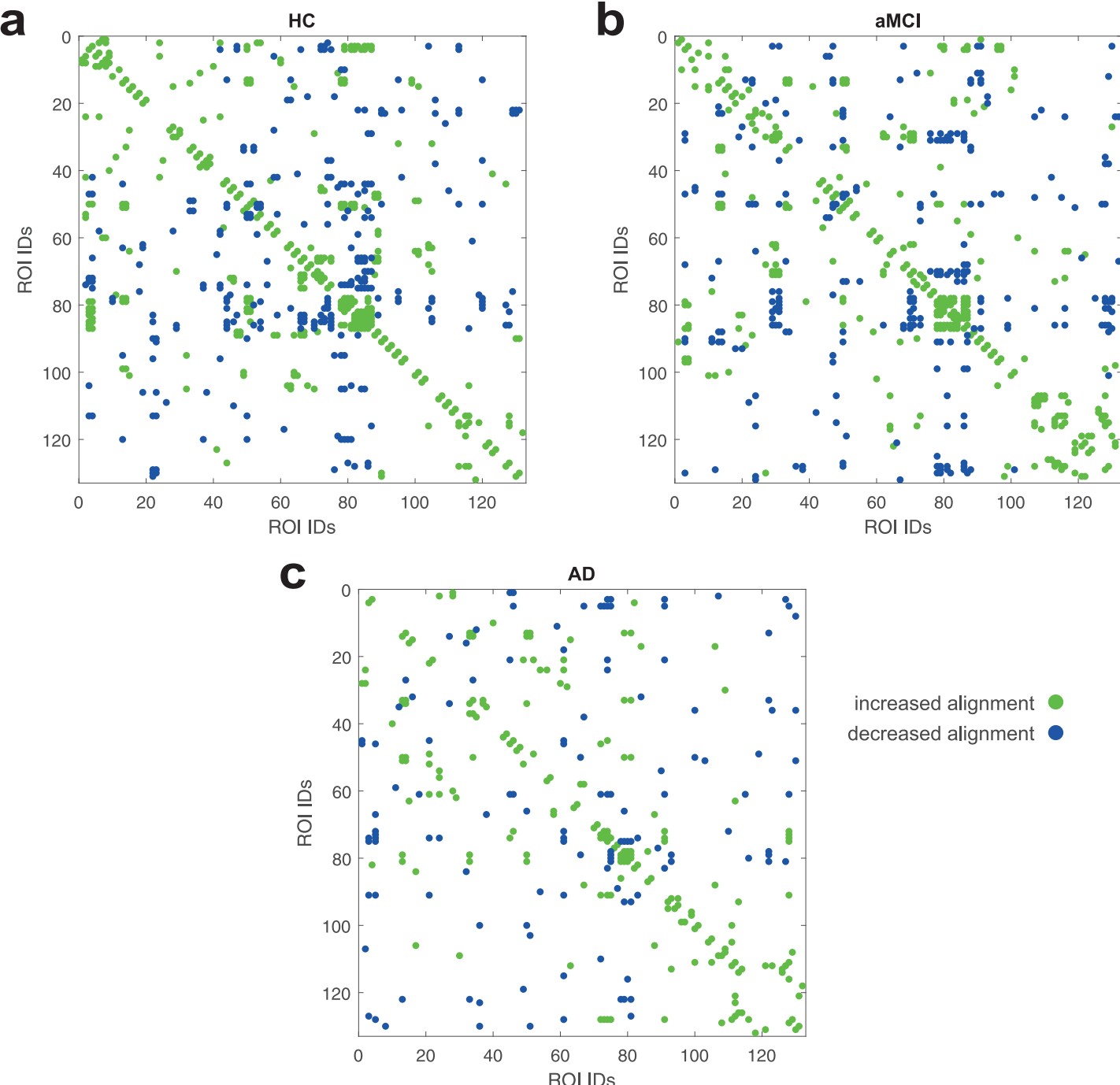

**Fig 4. Significant eigenvector alignment changes ($p \leq 2.5 \times 10^{-3}$) between a the HC, b the aMCI and c the AD groups versus random functional connectivity model.** An increase or decrease in alignment are noted from the perspective of the group being compared to the random model.

Finally, as noted in the Filtering Results section, to mitigate against significance arising from noisy data the results in the following sections shall only include ROI pairs that consistently display a significant difference, $p \leq 0.05$, across all three comparisons with different sets of 1000 randomised connectivity matrices.

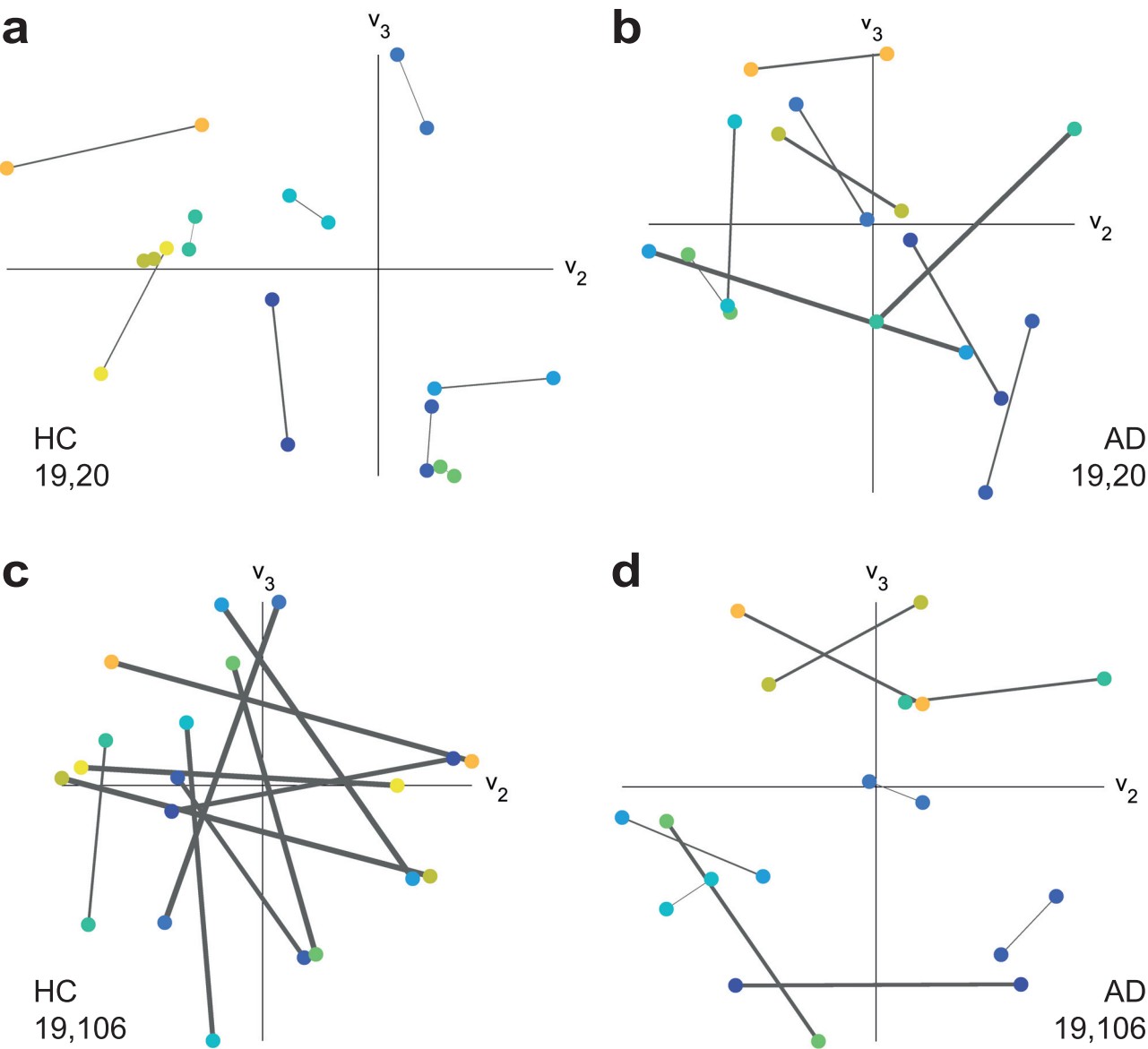

**Fig 5. ROI position comparison according to the 2ⁿᵈ and 3ʳᵈ dominant eigenvectors.** Each region is represented by a dot, with connections shown if regions belong to the same subject. The line thickness is proportional to the eigenvector alignment angle, assessed using the three dominant eigenvectors, where a larger thicknesses indicates a larger angle between ROIs. The right (19) and left (20) posterior superior temporal gyrus (pSTG) in **a** and **b**, with the right pSTG (19) and the brainstem (106) in **c** and **d**. HC subjects are shown in **a** and **c** with AD subjects in **b** and **d**.

## Visualising alignment

Two significant eigenvector alignment changes, from the AD-HC comparison, are highlighted in Fig 5 by displaying the ROIs in a Euclidean plane defined by the 2ⁿᵈ and 3ʳᵈ dominant eigenvectors ($v_2$ and $v_3$) of the adjacency matrix. In Fig 5a and 5b, the bilateral alignment of the posterior superior temporal gyrus (pSTG) is shown to be significantly lower ($p = 2 \times 10^{-3}$) for those with AD. This loss in alignment between homologous cortical areas in each hemisphere was expected given the findings presented in Fig 4. In Fig 5c and 5d, it is shown that this substantially lower alignment enables the most significant increase in alignment for the AD group ($p = 2 \times 10^{-5}$), between the right division of the pSTG and the brainstem.

Fig 5a details the close alignment of the left and right divisions of the pSTG, in HC subjects, with a mean alignment angle $\bar{\theta} = 0.29$ radians. This is in contrast to Fig 5b of the same two ROIs, in AD subjects, where the mean angle increases to $\bar{\theta} = 0.92$ rad.

The right pSTG is poorly aligned with the brainstem, for HC subjects, with a mean $\bar{\theta} = 1.60$ rad. This lack of alignment is visually clear, in Fig 5c, with the line between the ROIs crossing at least one axis and, for half of the subjects, crossing both axes. Again there is a contrast with the AD group where the mean angle decreases to $\bar{\theta} = 0.68$ rad for this alignment. This closer alignment for the AD group can be seen in Fig 5d where there are only a few lines crossing axes and one of those belongs to a pair of closely aligned ROIs near the origin.

Interestingly, the left pSTG also becomes significantly more aligned with the brainstem ($p = 9 \times 10^{-4}$), for the AD-HC comparison, despite having a decreased alignment with the right pSTG. This shift is significant bilaterally due to the notable lack of alignment in HC subjects between the pSTG and brainstem. The mean alignment angle, with respect to the brainstem, is $\bar{\theta} = 1.614$ rad for the left and $\bar{\theta} = 1.603$ rad for the right pSTG in the HC group. This becomes $\bar{\theta} = 0.888$ rad for the left and $\bar{\theta} = 0.682$ rad for the right in the AD group.

The brainstem is among one of the most aligned ROIs for the right pSTG in the AD group. The only more aligned ROIs, for the right pSTG, are the left cuneus ($\bar{\theta} = 0.649$ rad) and the left supracalcarine cortex ($\bar{\theta} = 0.680$ rad). Whereas the brainstem is less notably aligned to the left pSTG, which is most closely aligned to the left temporo-occipital middle temporal gyrus ($\bar{\theta} = 0.682$ rad) and the left superior frontal gyrus ($\bar{\theta} = 0.763$ rad).

The increased separation of the left and right divisions of pSTG has been highlighted before in [33] where AD is noted to cause disruption of the commissural system connecting the bilateral temporal and parietal cortical areas. Given the reported commissural system disruption in parietal cortical areas [33], it is notable that a significantly reduced alignment is also seen between the left and right superior parietal lobules (SPL) in the AD versus HC comparison ($p = 0.026$). In AD subjects, the left SPL is most aligned with the left juxtapositional lobule ($\bar{\theta} = 0.575$ rad) and the left precentral gyrus ($\bar{\theta} = 0.655$ rad). While the right SPL is most aligned with the left anterior supramarginal gyrus ($\bar{\theta} = 0.603$ rad) and the frontal medial cortex ($\bar{\theta} = 0.678$ rad).

The aMCI group also displays significant changes for the right pSTG, versus the HC group, with a significant decrease in alignment with respect to the left pSTG ($p = 0.043$) and an increase in alignment with respect to the brainstem ($p = 0.038$). In both cases aMCI is at an intermediary stage between what is observed from the HC and AD group.

## Subject group comparisons

In Fig 6, significant alignment differences between the AD, aMCI, and HC subject groups are presented, with the ROI IDs detailed in S1 Table. The AD versus HC comparison, in Fig 6a, confirms an expected result of decreased alignment between left and right side of the same cortical area with 11 ROI pairs displayed along the matrix diagonal. This consistent degradation of the commissural system is unique to the AD-HC comparison, with Fig 6b and 6c both reporting significant decreases and increases in alignment for the ROIs along the diagonal. Other clusters of significant alignment shifts are also observed in each comparison.

The AD versus HC case, Fig 6a, details significant realignments for the ROIs 81–89, which includes the parietal operculum cortex, planum polare, Heschl's gyrus, planum temporale and supracalcarine cortex. This cluster of ROIs display consistent decreases in alignment to the juxtapositional lobule cortex (51) and the left planum polare (82). While also displaying

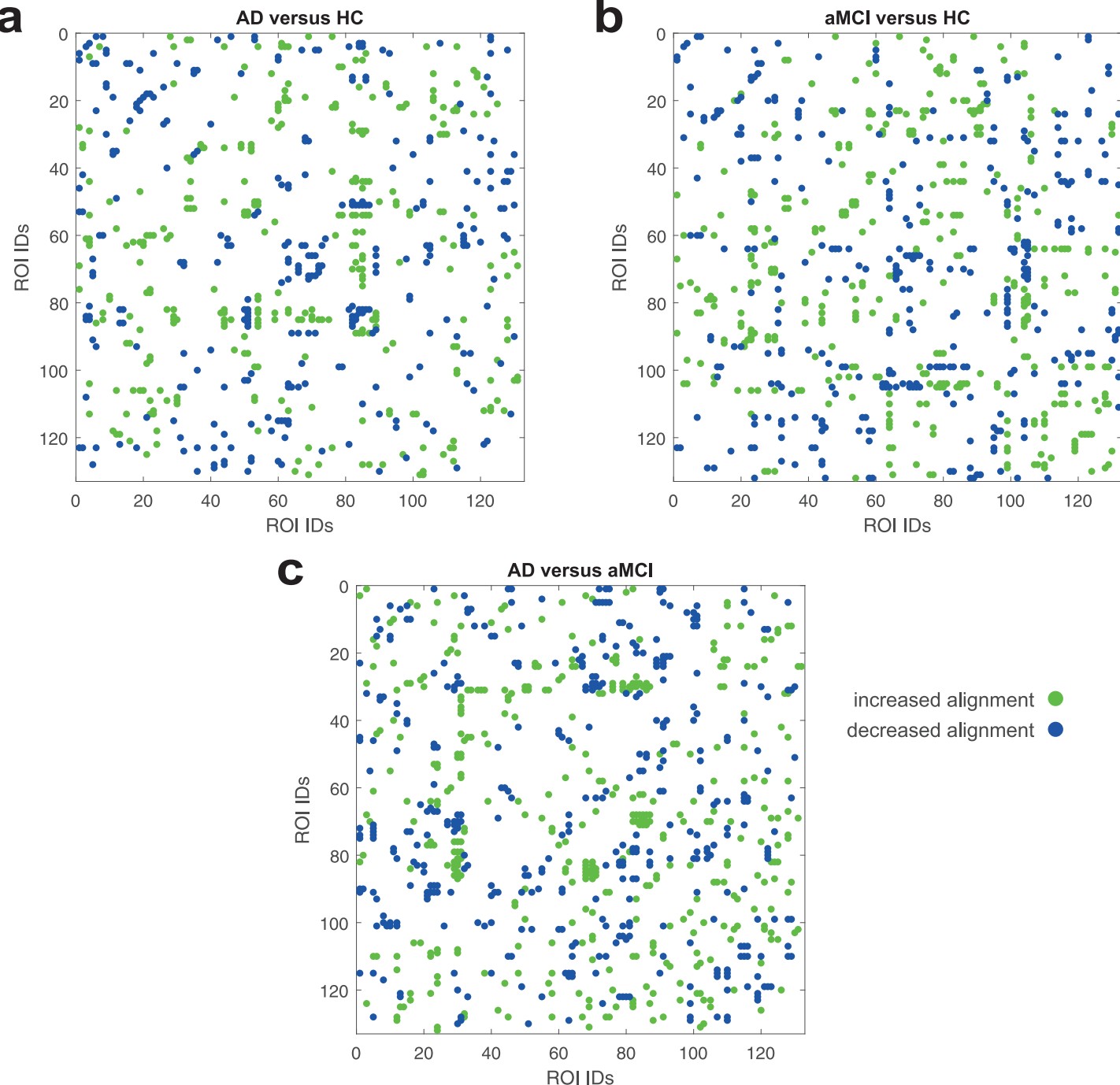

**Fig 6. Significant eigenvector alignment changes ($p \leq 0.05$) between a the AD and HC groups, b the AD and aMCI groups, and c the aMCI and HC groups.**
Significance only assessed for alignment pairs that also achieve $p \leq 0.05$ against the set of random models. An increase or decrease in alignment are noted from the perspective of the first group being compared to the second.

consistent increases in alignment to the right inferior temporal gyrus (29), left superior lateral occipital cortex (44), left paracingulate gyrus (54) and left supracalcarine cortex (89).

The aMCI versus HC case, in Fig 6b, displays significant disruption to the nucleus accumbens bilaterally (IDs 104 and 105). This disruption includes decreased alignment with ROIs

62–64 (parahippocampal gyrus) and 68–73 (the temporal fusiform and temporal occipital fusiform cortices). The nucleus accumbens has increased alignment with a number of ROIs in the notable 81–89 cluster, which again experience widespread changes in alignment including decreased alignment to the left pallidum (99).

In the AD versus aMCI case, Fig 6c, the inferior temporal gyrus (27–32) undergoes significant realignment, including decreases in alignment to the posterior temporal fusiform cortex (70–71) and increases in alignment with the aforementioned ROIs 81–89. The 81–89 cluster also sees widespread increases in alignment with the right temporal fusiform cortex (ID 68 and 70). In the following sections, we will highlight in more detail some of the notable findings from these comparisons.

## Alzheimer's disease

Table 1 details the ROIs with the largest number of significant changes in alignment between the AD and HC groups. This number does not distinguish between an increase or decrease in alignment, as both are present for those listed in Table 1. The left Heschl's gyrus and the bilateral planum polare are prominent in Table 1 and are part of the ROI 81–89 cluster highlighted in Fig 6a. It is notable that these two ROIs are in close alignment for HC subjects but become significantly less aligned in the AD group. Both sides of the Heschl's gyrus have a significant reduction in alignment with respect to the planum polare, but there is a greater change in mean alignment for the left Heschl's gyrus when compared with the left ($\Delta\bar{\theta} = 0.97$ rad, $p = 0.008$) and right planum polare ($\Delta\bar{\theta} = 0.91$ rad, $p = 0.007$).

It is interesting to note that despite the links between eigenvector alignment and eigenvector centrality (EC), the ROIs in Table 1 are not the same as those that show the most significant changes in EC. For the AD-HC group comparison, a significant decrease in EC is observed for the right posterior middle temporal gyrus ($p = 0.027$), left superior parietal lobule ($p = 0.049$), left juxtapositional lobule ($p = 0.044$) and anterior cingulate gyrus ($p = 0.016$). Alongside an increase in EC for the right caudate ($p = 0.012$), left cerebellum 4 & 5 ($p = 0.030$), and vermis 10 ($p = 0.001$).

## Amnestic mild cognitive impairment

Table 2 displays the ROIs with the largest number of significant changes in alignment between the aMCI and HC groups. The right posterior middle temporal gyrus is noteworthy for also having a significant change in its eigenvector centrality (EC) between the aMCI and HC groups. Significant EC decreases are observed in right posterior middle temporal gyrus ($p = 0.024$), vermis 7 ($p = 0.043$) and vermis 10 ($p = 0.012$). Significant EC increases are

**Table 1. For the AD comparison with HC (AD-HC), the ROIs with the largest number of significant eigenvector alignment changes ($p \leq 0.05$), with respect to other ROI, are detailed.**

| ID | ROI | AD-HC |
|---|---|---|
| 85 | Heschl's gyrus Left | 24 |
| 63 | Parahippocampal gyrus, anterior Left | 16 |
| 82 | Planum polare Right | 15 |
| 4 | Insular cortex Left | 13 |
| 54 | Paracingulate gyrus Left | 13 |
| 44 | Lateral occipital cortex, superior Left | 12 |
| 83 | Planum polare Left | 12 |
| 123 | Cerebellum 10 Left | 12 |

**Table 2. For the aMCI comparison with HC (aMCI-HC), the ROIs with the largest number of significant eigenvector alignment changes ($p \leq 0.05$), with respect to other ROI, are detailed.**

| ID | ROI | MCI-HC |
|---|---|---|
| 64 | Parahippocampal gyrus, posterior Right | 22 |
| 99 | Pallidum Left | 22 |
| 104 | Accumbens Right | 21 |
| 23 | Middle temporal gyrus, posterior Right | 19 |
| 105 | Accumbens Left | 16 |
| 24 | Middle temporal gyrus, posterior Left | 13 |
| 30 | Inferior temporal gyrus, posterior Left | 13 |
| 44 | Lateral occipital cortex, superior Left | 13 |

observed in the right posterior ($p = 0.042$) and left temporooccipital inferior temporal gyrus ($p = 0.020$), right occipital fusiform gyrus ($p = 0.047$), and right cerebellum 8 ($p = 0.021$).

## AD and aMCI comparison

Table 3 compares the AD and aMCI groups on the largest number of significant alignment changes, and displays that more ROIs experience widespread changes to their alignment in this case than the AD-HC or aMCI-HC comparisons.

The left planum polare experiences significant changes according to both EA and EC metrics. Significant decreases in EC are seen for the anterior cingulate cortex ($p = 0.044$) and the left planum polare ($p = 0.043$), while a significant increase in EC is noted for the left anterior supramarginal gyrus ($p = 0.026$).

## Hippocampal regions

The parahippocampal gyrus displays some notable changes in eigenvector alignment, which are captured in Table 4. In [34], the posterior parts of the parahippocampal gyri (pPHG) were found to be preferentially affected in age related memory decline. This is noted in [34] to

**Table 3. For the AD comparison with aMCI (AD-aMCI), the ROIs with the largest number of significant eigenvector alignment changes ($p \leq 0.05$), with respect to other ROI, are detailed.**

| ID | ROI | AD-MCI |
|---|---|---|
| 30 | Inferior temporal gyrus, posterior Left | 21 |
| 31 | Inferior temporal gyrus, temporooccipital Right | 20 |
| 110 | Cerebellum crus2 Right | 18 |
| 91 | Occipital pole Left | 17 |
| 70 | Temporal fusiform cortex, posterior Right | 16 |
| 24 | Middle temporal gyrus, posterior Left | 15 |
| 29 | Inferior temporal gyrus, posterior Right | 15 |
| 83 | Planum polare Left | 15 |
| 82 | Planum polare Right | 14 |
| 101 | Hippocampus Left | 14 |
| 115 | Cerebellum 6 Left | 14 |
| 5 | Superior frontal gyrus Right | 13 |
| 128 | Vermis 6 | 13 |
| 68 | Temporal fusiform cortex, anterior Right | 12 |
| 79 | Central opercular cortex Left | 12 |

**Table 4. Hippocampal regions: The number of significant changes in eigenvector alignment ($p \leq 0.05$) for each ROI with respect to other ROI are detailed for AD subjects compared to HC (AD-HC) as well as aMCI subjects compared to HC (aMCI-HC) and AD subjects compared to aMCI (AD-aMCI).**

| ID | ROI | AD-HC | aMCI-HC | AD-aMCI |
|----|-----|-------|---------|---------|
| 62 | Parahippocampal gyrus, anterior Right | 7 | 4 | 5 |
| 63 | Parahippocampal gyrus, anterior Left | 16 | 4 | 8 |
| 64 | Parahippocampal gyrus, posterior Right | 2 | 22 | 11 |
| 65 | Parahippocampal gyrus, posterior Left | 1 | 5 | 3 |
| 100 | Hippocampus Right | 2 | 2 | 8 |
| 101 | Hippocampus Left | 2 | 7 | 14 |

contrast with AD, where the anterior of the parahippocampal gyri (aPHG) has been found to be more severely affected [35, 36]. This pathology is supported by the results in Table 4, which reports a higher number of significant changes in eigenvector alignment for the left and right aPHG, in the AD-HC comparison with 16 and 7 respectively. While the largest change for the left and right pPHG is seen in the aMCI-HC comparison with 22 and 5 respectively. Table 4 also reveals that there are only a few significant changes in eigenvector alignment for the pPHG in the AD-HC case, despite the high variation in the aMCI-HC case.

For the hippocampus, Table 4 shows that the greatest change in alignment occurs between AD and aMCI subjects. In the aMCI group the left hippocampus is in closest alignment (according to the mean alignment angle) with the left, pars triangularis, inferior frontal Gyrus and the right hippocampus is closest to the left temporal pole. These alignments significantly decrease in AD subjects, with the left hippocampus in closest alignment to the right hippocampus while the right hippocampus is closest to the left amygdala.

## Default mode network

It has been repeatedly observed that the Default Mode Network (DMN) is affected by AD and aMCI with the posterior cingulate gyrus and precuneus often the focus of such studies [37]. We find that ROIs in the DMN are repeatedly highlighted by eigenvector centrality (EC) in each of the comparisons in this paper; decreased EC for the anterior cingulate in both the AD versus HC ($p = 0.018$) and AD versus aMCI ($p = 0.046$), decreased EC for the right angular gyrus in the aMCI versus HC comparison ($p = 0.025$) and increased EC for the left anterior supramarginal gyrus in AD versus aMCI ($p = 0.024$). In contrast, these regions do not show a large number of significant changes in eigenvector alignment, see Table 5. The main exception

**Table 5. Default Mode Network and related ROI: The number of significant changes in eigenvector alignment ($p \leq 0.05$) for each ROI with respect to other ROI are detailed for AD subjects compared to HC (AD-HC) as well as aMCI subjects compared to HC (aMCI-HC) and AD subjects compared to aMCI (AD-aMCI).**

| ID | ROI | AD-HC | aMCI-HC | AD-aMCI |
|----|-----|-------|---------|---------|
| 37 | Supramarginal gyrus, anterior Right | 2 | 5 | 1 |
| 38 | Supramarginal gyrus, anterior Left | 1 | 3 | 5 |
| 39 | Supramarginal gyrus, posterior Right | 0 | 5 | 2 |
| 40 | Supramarginal gyrus, posterior Left | 2 | 1 | 6 |
| 41 | Angular gyrus Right | 5 | 3 | 2 |
| 42 | Angular gyrus Left | 2 | 5 | 4 |
| 53 | Paracingulate gyrus Right | 8 | 5 | 2 |
| 54 | Paracingulate gyrus Left | 13 | 9 | 2 |
| 55 | Cingulate gyrus, anterior | 0 | 3 | 6 |
| 56 | Cingulate gyrus, posterior | 1 | 4 | 1 |
| 57 | Precuneous cortex | 1 | 2 | 4 |

is the left paracingulate gyrus for both the AD group and aMCI group when compared with HC subjects.

To explore this lack of alignment change, the connectivity matrices of the HC group are adapted by replacing the functional connectivity of one ROI. Specifically, the posterior cingulate gyrus (pCG) was selected as a prominent region in the DMN where we would have expected to see change due to AD. The pCG was taken from nine AD subjects and substituted into nine of the HC subjects. By comparing these altered HC connectivity matrices with those of the original HC group, more significant alignment shifts are seen than in the AD-HC comparison. For the subjects with a substituted pCG, there are now 8 significant changes in eigenvector alignment, whereas in the AD versus HC comparison there was only 1. Therefore, the changes to pCG's functional connectivity in isolation are notable but these are masked by other functional connectivity changes in AD subjects. For the substituted case, the pCG has decreased alignment to the left angular gyrus, left paracingulate gyrus and precuneus cortex and increased alignment to the left cuneal cortex, the bilateral lingual gyrus and the bilateral occipital fusiform gyrus.

This ROI substitution can not be detected as easily when only considering EC. There is a decrease in the mean EC value for the pCG ROI, but this is not significant when comparing the alterated HC connectivity matrices with HC subjects nor does it produce any significant changes with respect to other ROI.

In contrast to pCG, performing the same substitution process with the left Heschl's gyrus (which experiences 24 significant alignment changes between AD and HC) only 19 significant alignment changes are observed, where the majority overlap with those seen in the AD-HC comparison. Therefore, unlike the pCG the alignment change is amplified for the Heschl's gyrus by the other functional connectivity changes in AD.

Finally, applying the substitution approach for the anterior cingulate gyrus does not increase the number of significant alignment changes from 0. Therefore, in the following section we will explore another aspect of why these DMN ROI do not present with many significant alignment changes.

## Functional connectivity

In the Eigenvector Alignment section the difference between assessing changes in functional connectivity and EA were explored. These differences can result in ROIs experiencing a large number of significant changes in functional connectivity without these translating into significant changes in alignment. For instance by assessing functional connectivity changes for pairs of ROI in the AD versus HC case, the greatest number of changes were seen for the right planum polare and right pallidum with 14 significant changes each. The right planum polare is highlighted by EA in the Alzheimer's Disease section, but the right pallidum is a notable omission with only 3 significant realignments. This appears to be due, at least in part, to its lack of consistently high or low alignments to any ROIs, where the standard deviation of all the mean alignments for the right pallidum is $\sigma = 0.163$ in the HC group and $\sigma = 0.160$ in the AD group. This compares with the alignments of the right planum polare with $\sigma = 0.294$ in HC and $\sigma = 0.226$ in AD. The anterior cingulate gyrus was noted in the Default Mode Network for not experiencing many significant alignment changes, but like the right pallidum it does display a high number of significant functional connectivity changes, with 11 recorded. Again there is a lack of consistently high or low alignments to any ROI, in either HC or AD subjects ($\sigma = 0.184$ for HC and $\sigma = 0.168$ for AD) and without consistently clear alignments few significant changes in alignment will be detected.

## Discussion

The use of CDI to compare community size between subject groups at either end of the influence spectrum, rather than one of the plethora of other community detection methods, is justified through its explicit linkage of network division with global network influence. Furthermore, it is worth highlighting that CDI is deterministic with no stochastic processes that are are commonly found in community detection algorithms but can reduce the reliability of findings, such as those presented herein on community size. Eigenvector alignment also links network division and influence in a similar manner to CDI and hence the clear differences in community size, found with CDI after thresholding with CST, supports the application of EA with CST on this dataset.

The insights gained from EA are not exclusive to using three dominant eigenvectors. The Introduction section notes that a range of eigenvectors have been employed previously to identify neuronal circuitry. The use of the most dominant eigenvectors simply ensures that the changes exposed are amongst the most important in terms of information flow around the brain, as captured by functional connectivity. The use of three eigenvectors is supported in [12] for identifying influential leaders in systems governed by consensus dynamics. Our findings appear to support the use of three eigenvectors, with AD subjects most clearly identified from the size of their most and least influential communities of dynamical influence (CDI) when using three dominant eigenvectors.

When comparing EA between the HC, aMCI, and AD groups a filter was applied to only include ROI pairs that showed a significant difference in alignment to randomly generated connectivity matrices. This filter was intended to mitigate against false detections of significance given the small size of the dataset. This is potentially good practice even on larger datasets, but it may be worth exploring if this obscures any genuine alignment changes.

In Fig 4 significant disruption to the commissural system of subjects with AD is noted. We also report on restructuring in response to this disruption, where in Fig 5 either side of posterior superior temporal gyrus (pSTG) becomes increasingly separated, in terms of alignment, as a person progresses from HC → aMCI → AD. This separation results in a very significant increase in alignment to the brainstem ROI for both sides of the pSTG, which appears to be compensatory restructuring of the brain's connections in AD. Especially for the right pSTG where the brainstem transitions from being one of the least aligned ROIs in HC subjects to one of the most aligned in AD subjects. A recent investigation using electrophysiological methods indicated hypersensitivity within the brainstem in people with MCI, which may indicate a compensatory process [38].

It is acknowledged that ROIs not strongly associated with the AD pathology experience a large number of significant alignment changes—such as the parietal operculum cortex, planum polare, Heschl's gyrus, and planum temporale—whilst other regions that are known to lose functional connectivity maintain a similar alignment to that seen in Healthy Control subjects—such as the ROIs in the Default Mode Network (DMN). The finding of significant disruption to left Heschl's gyrus (also referred to as the transverse temporal gyrus) is interesting as there is growing evidence of AD associated changes in the auditory cortex, which includes the pSTG and Heschl's gyrus. The volume of left Heschl's gyrus can discriminate people with AD from healthy controls (sensitivity 86.5%; specificity 79.7%) [39]. Functional connectivity changes in the Heschl's gyrus are associated with age-related hearing loss [40], which is a known risk factor for Alzheimer's disease [41]. Furthermore, in [42], the right auditory cortex is proposed to be initially more resistant to degenerative changes than the left auditory cortex. This conclusion was drawn from the finding that ipsilateral auditory processing is delayed only in the left side for AD subjects. Given the findings in [42], it is worth noting the

asymmetry in alignment changes for the two sides of the Heschl's gyrus, with 24 significant changes for the left and 9 for the right in the AD-HC case. This auditory asymmetry has also been noted in [43] where significant differences in connectivity between the left auditory cortex and the posterior hippocampus is found in carriers of the APOE$\epsilon$4 protein, which represents a high risk of AD diagnosis. Our findings relating to the pSTG and Heschl's gyrus contribute to accumulating evidence that, during the course of AD, the auditory cortex appears to be undergoing substantial changes in volume, connectivity and influence.

It is well-established that the DMN experiences reduced functional connectivity in those with AD [37, 44, 45], therefore changes in eigenvector centrality (EC) and alignment may be anticipated. Differences in EC are seen for the DMN including a clear decrease in EC for the anterior cingulate gyrus (aCG) in AD with respect to both HC and aMCI groups. But the same was not seen for eigenvector alignment, with two reasons highlighted for this. Firstly, eigenvector alignment assesses relative changes in the connectivity network with significant changes in functional connectivity no guarantee of alignment shifts, especially when there are widespread connectivity changes. This claim is supported by replacing the posterior cingulate gyrus (pCG) of HC subjects, with that of AD subjects, to demonstrate that the pCG's functional connectivity changes in isolation would produce significant shifts in eigenvector alignment. Another contributing factor is a lack of consistently high or low alignments to any ROI, as is the case for the aCG. This lack of consistency ensures that significant functional connectivity changes do not convert into significant alignment shifts.

This set of analyses has shown that eigenvector alignment can offer insights into the network properties of the brain, and may be usefully applied in conjunction with other methods to gain a fuller picture of the brain's function. Even within eigenvector alignment, there are a number of elements that can be combined or explored separately. For example, the number of significant alignment changes for a ROI may be few, but these changes might be large in magnitude and extremely important for brain function (e.g., a cognitive ability could be impaired through a disconnection between two single ROIs).

## Conclusions

The embedding of functional connectivity matrices, in Euclidean space defined by the system's dominant eigenvectors, provides new insights into structural network changes occurring in subjects with AD and aMCI. Differences in the network community structure of AD and HC subjects can be discerned by comparing their communities of dynamical influence. But more specific insights can be identified by assessing eigenvector alignment, introduced here as a method for comparing individual ROI position vectors in this eigenvector defined coordinate system. EA is shown to provide a holistic assessment of how functional connectivity changes affect the relationship between any two ROI. EA produces expected results in comparisons with randomly generated matrices, where ROI pairs from either side of the same cortical area present as some of the most consistently, closely, aligned pairs. From these comparisons and those between AD and HC subjects, we can see that there is a degradation of the commissural system in AD that displays as a loss of alignment for these same cortical area pairs. For ROIs, such as the posterior superior temporal gyrus (pSTG), compensatory activity can also be detected with EA through the emergence of new consistently close alignments. Eigenvector centrality results are found to often be distinct from those of EA, with the ROIs in the Default Mode Network (DMN) a clear example where significant changes in functional connectivity and EC do not result in EA shifts. The lack of alignment changes highlight that loss in connectivity does not guarantee structural changes, as widespread connectivity loss could result in decreased flow through the network but no structural change. EA provides a mechanism for

identifying these purely structural network changes and as such highlights ROIs not commonly associated with the AD pathology. This includes the auditory cortex, with the posterior superior temporal gyrus, Heschl's gyrus, and planum polare undertaking some of the most significant changes in alignment for those with AD and also experiencing notable changes in those with aMCI. EA both detects changes in aMCI that are amplified in AD, such as the pSTG's shift from bilateral alignment to brainstem alignment. Alongside distinct changes in aMCI that differ starkly from those with AD, where prominent alignment shifts are seen only in the posterior parahippocampal gyrus in aMCI while in AD the anterior PHG is predominantly affected.

This analysis formed a proof of concept for eigenvector alignment, which demonstrates clear potential for wider application but would benefit from further analysis on larger datasets that can confirm the reliability of this method and its validity in identifying biomarkers of disease.

## Supporting information

**S1 Table. A list of the ROIs identified according to the CONN atlas.**
(PDF)

**S2 Table. Eigenvector alignment where the number of significant changes in eigenvector alignment for each ROI with respect to other ROI are detailed for the comparisons AD versus HC, aMCI versus HC, and AD versus aMCI.**
(PDF)

## Author Contributions

**Conceptualization:** Ruaridh A. Clark.

**Data curation:** Niia Nikolova.

**Formal analysis:** Ruaridh A. Clark.

**Methodology:** Ruaridh A. Clark, Niia Nikolova, Malcolm Macdonald.

**Supervision:** William J. McGeown, Malcolm Macdonald.

**Visualization:** Ruaridh A. Clark.

**Writing – original draft:** Ruaridh A. Clark, Niia Nikolova.

**Writing – review & editing:** Niia Nikolova, William J. McGeown, Malcolm Macdonald.

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
