## [Decision Letter · Decision Letter 0]

2 Jun 2020

PONE-D-20-08029

Eigenvector alignment: assessing functional network changes in amnestic mild cognitive impairment and Alzheimer's disease

PLOS ONE

Dear Dr. Clark,

Thank you for submitting your manuscript to PLOS ONE. After careful consideration, we feel that it has merit but does not fully meet PLOS ONE’s publication criteria as it currently stands. Therefore, we invite you to submit a revised version of the manuscript that addresses the points raised during the review process.

We look forward to receiving your revised manuscript.

Kind regards,

Hocine Cherifi

Academic Editor

PLOS ONE

2. Please ensure that you refer to Figure 2 in your text as, if accepted, production will need this reference to link the reader to the figure.

Reviewers' comments:

Reviewer's Responses to Questions

**Comments to the Author**

1. Is the manuscript technically sound, and do the data support the conclusions?

Reviewer #1: Yes

Reviewer #2: Partly

2. Has the statistical analysis been performed appropriately and rigorously? 

Reviewer #1: Yes

Reviewer #2: Yes

3. Have the authors made all data underlying the findings in their manuscript fully available?

Reviewer #1: Yes

Reviewer #2: Yes

4. Is the manuscript presented in an intelligible fashion and written in standard English?

Reviewer #1: Yes

Reviewer #2: Yes

5. Review Comments to the Author

Reviewer #1: I think the idea of the study is very meaningful and interesting. Research design is appropiated and described fully. Results and conclusions are presented clearly. However, it would be useful for the reader to receive more information regarding the participants (demographic data, educational level, socioeconomic level, cognitive profile, neuropsychiatric symptoms).

Reviewer #2: The authors present a novel and potentially interesting approach to network analysis in functional connectivity analysis of neuroimaging data. However, the technique is applied only in a single very small dataset. There is a wealth of available open access datasets this method could be applied to. For example, the ADNI dataset would be particularly relevant for this paper. It is entirely reasonable to develop the dataset in a small test dataset, then go on to test the validity of the technique in another dataset.

I do hope the authors would consider doing this, given the potentially interesting findings. However, in its current form it would be rash to accept eigenvector allgnment as a reliable method for this kind of analysis.

In addition, the authors need to demonstrate the the CDI is better than some of the other methods outlined in the introduction for detecting communities. One issue with community detection is the variability depending on choice of parameters (eg thresholding) and stochastic effects. The authors could consider if they can conclusively demonstrate the superiority of CDI in this respect.

6. PLOS authors have the option to publish the peer review history of their article (what does this mean?). If published, this will include your full peer review and any attached files.

Reviewer #1: No

Reviewer #2: Yes: Timothy Rittman

---

## [Author Response · Author response to Decision Letter 0]

25 Jun 2020

Additional Requirements:

- File links do not appear to work, but followed guidance on PLOS ONE website.

2. Please ensure that you refer to Figure 2 in your text as, if accepted, production will need this reference to link the reader to the figure.

- Figure 2 is now referred to in text.

- Supporting information captions included.

Review Comments to the Author:

Reviewer #1: I think the idea of the study is very meaningful and interesting. Research design is appropiated and described fully. Results and conclusions are presented clearly. However, it would be useful for the reader to receive more information regarding the participants (demographic data, educational level, socioeconomic level, cognitive profile, neuropsychiatric symptoms).

Response: Added additional information on participants in the Dataset subsection:

“Probable AD diagnosis was defined by NINCDS-ADRDA consensus criteria [1], with a general cognitive evaluation made using Mini-Mental State Examination (MMSE). The mean MMSE score was 21.5 (SD 3.7) for the AD group, 25.8 (SD 2.3) for the MCI group and 29.3 (SD 0.67) for the HC group. The mean age of the AD group was 72.3 (SD 8.3), the mean of the MCI group was 70.7 (SD 7.1) and the mean of the HC group was 66.0 (SD 9.6). The mean education was 8.6 (SD 3.6) in the AD group, 11.1 (SD 3.5) in the MCI group and 14.5 (SD 3.0) in the HC group. There were 6 females in the AD group, 4 in the MCI group and 3 in the HC group. For additional details on the participants in the dataset, see [2].”

[1] G. McKhann, D. Drachman, M. Folstein, R. Katzman, D. Price, and E. M. Stadlan, ‘Clinical diagnosis of Alzheimer’s disease: report of the NINCDS-ADRDA Work Group under the auspices of Department of Health and Human Services Task Force on Alzheimer’s Disease’, Neurology, vol. 34, no. 7, pp. 939–944, Jul. 1984, doi: 10.1212/wnl.34.7.939.

[2] D. Mascali et al., ‘Intrinsic Patterns of Coupling between Correlation and Amplitude of Low-Frequency fMRI Fluctuations Are Disrupted in Degenerative Dementia Mainly due to Functional Disconnection’, PLOS ONE, vol. 10, no. 4, p. e0120988, Apr. 2015, doi: 10.1371/journal.pone.0120988.

Additional information on neuropsychiatric symptoms, specific cognitive symptomology and socioeconomic status was not available for inclusion.

Reviewer #2: The authors present a novel and potentially interesting approach to network analysis in functional connectivity analysis of neuroimaging data. However, the technique is applied only in a single very small dataset. There is a wealth of available open access datasets this method could be applied to. For example, the ADNI dataset would be particularly relevant for this paper. It is entirely reasonable to develop the dataset in a small test dataset, then go on to test the validity of the technique in another dataset.

I do hope the authors would consider doing this, given the potentially interesting findings. However, in its current form it would be rash to accept eigenvector allgnment as a reliable method for this kind of analysis.

Response: We completely agree that there is a lot of potential for applying our methods to larger datasets to both validate the results presented in the paper and explore the reliability of these results as biomarkers for disease. This is work we aspire to complete following this publication and we have adjusted the conclusion to reflect these points and make the provisional nature of these findings clearer:

“This analysis formed a proof of concept for eigenvector alignment, which demonstrates clear potential for wider application but would benefit from further analysis on larger datasets that can confirm the reliability of this method and its validity in identifying biomarkers of disease.”

In addition to this clarification, we have made every effort to increase the accessibility of our work by making the eigenvector alignment algorithm, relevant scripts and processed dataset available at [3]. In this way, we provide anyone with the tools to validate our work in this paper and investigate the performance on other datasets.

[3] Clark, R. (2020, June 10). Eigenvector Alignment (Version v1.0). Zenodo. http://doi.org/10.5281/zenodo.3888075

Reviewer #2: In addition, the authors need to demonstrate the the CDI is better than some of the other methods outlined in the introduction for detecting communities. One issue with community detection is the variability depending on choice of parameters (eg thresholding) and stochastic effects. The authors could consider if they can conclusively demonstrate the superiority of CDI in this respect.

Response: Firstly, CDI is selected as it forms communities around the most influential network nodes, which creates an explicit link between the defined communities and the influence of nodes in the network. This link allows communities to be categorised as most or least influential and, hence, without this link the community size comparison would not be possible. We have attempted to clarify this point throughout the paper:

Text added in the Communities of dynamical influence section:

“While many other community detection algorithms exist, CDI is the focus here as it explicitly connects network influence with community designation. It is also worth noting that, unlike many other community detection methods, CDI is deterministic and not susceptible to stochastic processes changing community designations.

Text added in the Discussion section:

“The use of CDI for this analysis, rather than one of the plethora of other community detection methods, is justified through its explicit linkage of network division with global network influence, which is also fundamental to eigenvector alignment. Furthermore, it is worth highlighting that CDI is deterministic with no stochastic processes that are commonly found in community detection algorithms but can reduce the reliability of findings, such as those presented herein on community size.”

The reviewer has raised valid concerns around the issues of stochastic effects, thresholding, and other factors affecting the community designation. We have attempted to address these concerns by first noting that CDI is deterministic as is now highlighted in the previously quoted text. However, thresholding and other factors can influence CDI’s community designations. Hence, we have made efforts to add further analysis and discussion to support the findings of differences in community size. The additional analysis includes an extension to the influential community size comparison, where significant differences in the mean sizes of the two least and the two most influential communities are found. The analysis on the use of arbitrary thresholds, instead of the Cluster Span Threshold, has also been updated with significant results seen for certain threshold values and the community size relationship between groups remaining consistent:

“The pattern of AD subjects having larger most influential and smaller least influential communities when compared with HC subjects is resilient to threshold variation.”

The CDI algorithm settings that could influence community designation are now highlighted in the text as the number of input eigenvectors and the choice of eigenvector scaling. Both of these aspects are analysed and their impact on the significance of the findings highlighted, while the community size pattern continues to be resilient to variation this time from the algorithm settings.

Finally, this analysis is replicable with the code used to identify communities of dynamical influence made available and referenced in the paper at [4].

[4] Clark R., Communities of Dynamical Influence (Version v1.0); 2020. Available from:https://doi.org/10.5281/zenodo.3878931

---

## [Decision Letter · Decision Letter 1]

15 Jul 2020

PONE-D-20-08029R1

Eigenvector alignment: assessing functional network changes in amnestic mild cognitive impairment and Alzheimer's disease

PLOS ONE

Dear Dr. Clark,

Thank you for submitting your manuscript to PLOS ONE. After careful consideration, we feel that it has merit but does not fully meet PLOS ONE’s publication criteria as it currently stands. Therefore, we invite you to submit a revised version of the manuscript that addresses the points raised during the review process.

We look forward to receiving your revised manuscript.

Kind regards,

Hocine Cherifi

Academic Editor

PLOS ONE

Additional Editor Comments (if provided):

As suggested by the reviewer, the work will gain in momentum by putting it more clearly in context.

Reviewers' comments:

Reviewer's Responses to Questions

**Comments to the Author**

1. If the authors have adequately addressed your comments raised in a previous round of review and you feel that this manuscript is now acceptable for publication, you may indicate that here to bypass the “Comments to the Author” section, enter your conflict of interest statement in the “Confidential to Editor” section, and submit your "Accept" recommendation.

Reviewer #1: All comments have been addressed

Reviewer #2: (No Response)

2. Is the manuscript technically sound, and do the data support the conclusions?

Reviewer #1: Yes

Reviewer #2: Partly

3. Has the statistical analysis been performed appropriately and rigorously? 

Reviewer #1: Yes

Reviewer #2: Yes

4. Have the authors made all data underlying the findings in their manuscript fully available?

Reviewer #1: Yes

Reviewer #2: Yes

5. Is the manuscript presented in an intelligible fashion and written in standard English?

Reviewer #1: Yes

Reviewer #2: Yes

6. Review Comments to the Author

Reviewer #1: (No Response)

Reviewer #2: Many thanks for revising the manuscript, and in particular strengthening the methodological aspects of the paper. This is very much welcome. I appreciate the discussion of stochastic vs deterministic effect, this is much clearer.

I didn't perhaps explain my comments as well as I could have done previously. It's not clear to me whether this paper is 1. trying to demonstrate that this method can be applied in brain imaging, which is fairly trivial, 2. validating the method by reproducing a well known change found by another method, or, 3. proposing a new insight in to Alzheimer's disease. To help this, it would be useful to more clearly state a hypothesis to test.

The results as presented are therefore rather exploratory and a challenging to relate to what we know about AD. For example, Heschl's gyrus is not typically involved early in AD, and alignment of temporal lobe regions to the brainstem is difficult to explain. It is difficult to tell whether these hard to explain findings are an issue with the underlying method, the small sample size, or whether they represent a novel insight in to AD.

In order to assess the underlying method, it would be helpful to have a more clear hypothesis of expected findings based on previous work, or to compare with another method. I accept that comparing with other methods can be difficult when trying to demonstrate some superiority - ideally you'd like to show some overlap in the findings, but that the 'new' method demonstrates additional changes.

In order to clarify whether the small sample size is an issue, validating in a different dataset would be ideal. In many ways, it would be better to 'train' on the larger dataset and 'validate' on a smaller dataset. That's why I suggested ADNI which is readily available and has comparable data with functional imaging in AD, MCI and health control groups.

If the authors are trying to validate the method in neuroimaging in general, it would be better to choose to test an hypothesis that is more established rather than AD where there are conflicting reports, or to use modelled data such as with the virtual brain (https://www.thevirtualbrain.org/tvb/zwei).

7. PLOS authors have the option to publish the peer review history of their article (what does this mean?). If published, this will include your full peer review and any attached files.

Reviewer #1: No

Reviewer #2: **Yes: **Timothy Rittman

---

## [Author Response · Author response to Decision Letter 1]

6 Aug 2020

Reviewer #2: Many thanks for revising the manuscript, and in particular strengthening the methodological aspects of the paper. This is very much welcome. I appreciate the discussion of stochastic vs deterministic effect, this is much clearer.

I didn't perhaps explain my comments as well as I could have done previously. It's not clear to me whether this paper is 1. trying to demonstrate that this method can be applied in brain imaging, which is fairly trivial, 2. validating the method by reproducing a well known change found by another method, or, 3. proposing a new insight in to Alzheimer's disease. To help this, it would be useful to more clearly state a hypothesis to test.

The results as presented are therefore rather exploratory and a challenging to relate to what we know about AD. For example, Heschl's gyrus is not typically involved early in AD, and alignment of temporal lobe regions to the brainstem is difficult to explain. It is difficult to tell whether these hard to explain findings are an issue with the underlying method, the small sample size, or whether they represent a novel insight in to AD.

In order to assess the underlying method, it would be helpful to have a more clear hypothesis of expected findings based on previous work, or to compare with another method. I accept that comparing with other methods can be difficult when trying to demonstrate some superiority - ideally you'd like to show some overlap in the findings, but that the 'new' method demonstrates additional changes.

In order to clarify whether the small sample size is an issue, validating in a different dataset would be ideal. In many ways, it would be better to 'train' on the larger dataset and 'validate' on a smaller dataset. That's why I suggested ADNI which is readily available and has comparable data with functional imaging in AD, MCI and health control groups.

If the authors are trying to validate the method in neuroimaging in general, it would be better to choose to test an hypothesis that is more established rather than AD where there are conflicting reports, or to use modelled data such as with the virtual brain (https://www.thevirtualbrain.org/tvb/zwei).

Thank you for raising these concerns, I think in addressing your concerns we have significantly improved the clarity of the method and the reliability of the results. 

Authors' response: In response to what the paper is trying to achieve we believe it now fulfils all 3 of your criteria:

- We believe it achieves 1 that as you state is fairly trivial.

- We agree that it did not convincingly achieve 2 in the previous submission and have now introduced a comparison between subject groups and random models, which validates that eigenvector alignment functions as we’d expect with respect to the healthy control group in this dataset.

- We continue to propose insights that are not strongly associated with the AD pathology but have included further evidence from other studies that support these findings.

The issues highlighted with the paper appear to stem from a lack of clarity on how the underlying method functions and whether it produces reliable findings, especially on a small sample size. Starting with the underlying method, we have added a toy example in the methods section to illustrate that eigenvector alignment is essentially a comparison of pairwise functional connectivity. This toy example also highlights how our method’s holistic assessment of the changing relationship between two ROIs differs from just assessing differences in functional connectivity.

Taking the intuitions gained from the toy example, we have added a validation of eigenvector alignment on the current dataset by comparing each subject group with random models of connectivity matrices. The results of these comparisons consistently display increased alignment for ROI pairs from either side of the same cortical area. This is as expected, where the same cortical area would be in closer alignment in subject data than in randomised networks. The use of different sets of random models shows that eigenvector alignment is capable of consistent detection of significant ROI pairs.

Following on from this validation step, the random model analysis is also used to filter the results in the subject group comparisons. In this way, only ROI pairs that display significantly different alignment from the sets of random models are eligible for comparisons between subject groups. This filter does influence the results of the paper but not enough to affect the main findings of the paper and adds reassurance to the reader that we have mitigated against false detections of significance given our sample size.

Finally, we have added additional discussion around the changes seen in the brainstem and auditory cortex by highlighting how these correspond with other research on AD and memory impairment. Including:

“A recent investigation using electrophysiological methods indicated hypersensitivity within the brainstem in people with MCI, which may indicate a compensatory process [38].”

“Functional connectivity changes in the Heschl’s gyrus are associated with age-related hearing loss [40], which is a known risk factor for Alzheimer’s disease [41].”

---

## [Editor Report · Decision Letter 2]

12 Aug 2020

Eigenvector alignment: assessing functional network changes in amnestic mild cognitive impairment and Alzheimer's disease

PONE-D-20-08029R2

Dear Dr. Clark,

We’re pleased to inform you that your manuscript has been judged scientifically suitable for publication and will be formally accepted for publication once it meets all outstanding technical requirements.

Kind regards,

Hocine Cherifi

Academic Editor

PLOS ONE
---

## [Editor Report · Acceptance letter]

14 Aug 2020

PONE-D-20-08029R2 

Eigenvector alignment: assessing functional network changes in amnestic mild cognitive impairment and Alzheimer's disease 

Dear Dr. Clark:

I'm pleased to inform you that your manuscript has been deemed suitable for publication in PLOS ONE. Congratulations! Your manuscript is now with our production department. 

Kind regards, 

on behalf of

Professor Hocine Cherifi 

Academic Editor

PLOS ONE